# Proteomic and Lipidomic Profiling of Calves Experimentally Co-Infected with Influenza D Virus and *Mycoplasma bovis*: Insights into the Host–Pathogen Interactions

**DOI:** 10.3390/v16030361

**Published:** 2024-02-27

**Authors:** Ignacio Alvarez, Mariette Ducatez, Yongzhi Guo, Adrien Lion, Anna Widgren, Marc Dubourdeau, Vincent Baillif, Laure Saias, Siamak Zohari, Jonas Bergquist, Gilles Meyer, Jean-Francois Valarcher, Sara Hägglund

**Affiliations:** 1Division of Ruminant Medicine, Department of Clinical Sciences, Swedish University of Agriculture Sciences, 8 Almas Allé, 75007 Uppsala, Swedenjean-francois.valarcher@slu.se (J.-F.V.); sara.hagglund@slu.se (S.H.); 2IHAP, Université de Tolouse, INRAE, ENVT, 31076 Toulouse, France; 3Department of Chemistry-BMC, Analytical Chemistry and Neurochemistry, Uppsala University, Husargatan 3, 75124 Uppsala, Sweden; anna.widgren@kemi.uu.se (A.W.); jonas.bergquist@kemi.uu.se (J.B.); 4Ambiotis SAS, 3 Rue des Satellites, 31400 Toulouse, France; 5Department of Microbiology, Swedish Veterinary Agency, Ullsvägen 2B, 75189 Uppsala, Sweden; siamak.zohari@sva.se; 6Department of Animal Biosciences, Swedish University of Agricultural Sciences, Ulls väg 26, 75007 Uppsala, Sweden

**Keywords:** Influenza D, *Mycoplasma bovis*, proteomics, lipidomics, oxylipids, eicosanoids, bovine respiratory disease, cattle, co-infection

## Abstract

The role of Influenza D virus (IDV) in bovine respiratory disease remains unclear. An in vivo experiment resulted in increased clinical signs, lesions, and pathogen replication in calves co-infected with IDV and *Mycoplasma bovis* (*M*. *bovis*), compared to single-infected calves. The present study aimed to elucidate the host–pathogen interactions and profile the kinetics of lipid mediators in the airways of these calves. Bronchoalveolar lavage (BAL) samples collected at 2 days post-infection (dpi) were used for proteomic analyses by liquid chromatography-tandem mass spectrometry (LC-MS/MS). Additionally, lipidomic analyses were performed by LC-MS/MS on BAL samples collected at 2, 7 and 14 dpi. Whereas *M. bovis* induced the expression of proteins involved in fibrin formation, IDV co-infection counteracted this coagulation mechanism and downregulated other acute-phase response proteins, such as complement component 4 (C4) and plasminogen (PLG). The reduced inflammatory response against *M. bovis* likely resulted in increased *M. bovis* replication and delayed *M. bovis* clearance, which led to a significantly increased abundance of oxylipids in co-infected calves. The identified induced oxylipids mainly derived from arachidonic acid; were likely oxidized by COX-1, COX-2, and LOX-5; and peaked at 7 dpi. This paper presents the first characterization of BAL proteome and lipid mediator kinetics in response to IDV and *M. bovis* infection in cattle and raises hypotheses regarding how IDV acts as a co-pathogen in bovine respiratory disease.

## 1. Introduction

The increased use of molecular technologies has allowed for the discovery of previously unknown viruses, such as Influenza D virus (IDV), in the upper respiratory tract of cattle [1,2,3]. In addition, the widespread detection of high IDV-specific antibody titers in cattle on different continents suggests that cattle are the primary reservoir for IDV [4,5,6,7,8,9,10,11]. Although the precise role of IDV in bovine respiratory disease (BRD) remains unclear, several epidemiologic and molecular studies indicated that this virus is associated with respiratory clinical signs in calves [1,2,3,7,12]. In addition, experimental studies demonstrated that IDV induces respiratory clinical signs and replicates in both the upper and lower respiratory tract of calves [13,14]. To date, IDV has also been studied during co-infection with *S. aureus* in mice, *M. ovipneumoniae* in lambs, and *M. haemolytica* and *M. bovis* in cattle, with conflicting results [13,15,16,17]. In mice, a primary infection with IDV did not cause any signs of disease and did not increase the susceptibility to a secondary *S. aureus* infection. In lambs, IDV alone did not induce clinical signs, but previous exposure to *M. ovipneumoniae* enhanced the inflammatory response induced by IDV. Furthermore, Zhang et al. infected calves intranasally with IDV and five days later intratracheally with *M. haemolytica.* The co-infected calves developed fewer clinical signs and less severe lesions than those infected with *M. haemolytica* alone, suggesting that IDV can reduce disease as a co-pathogen [17].

Among the different co-infections studied so far, the interaction between IDV and *M. bovis* showed the most negative impact on the health of the animals. Compared with calves infected with IDV or *M. bovis* alone, the IDV and *M. bovis* co-infected calves developed more severe clinical signs and more extensive pathological lesions in the respiratory tract. In addition, IDV co-infection increased the replication of *M. bovis*. The interaction between IDV and *M. bovis* was additionally studied in ex vivo models using precision-cut lung slices by Gaudino et al. [18]. The results suggested that a primary IDV infection promoted *M. bovis* superinfection by increasing bacterial replication and by causing damage to lung pneumocytes. Additional experiments using cytosolic helicase and Toll-like receptor (TLR) agonists indicated that IDV increases the susceptibility of cattle to bacterial superinfections, such as *M. bovis*, by impairing the innate immune response [18].

As the interaction between IDV and *M*. *bovis* was demonstrated to be particularly detrimental, understanding the disease mechanisms involved in this process is crucial. Proteomic analysis provides insights into the immune response and allows us to identify proteins or pathways involved in host defense or pathogenic mechanisms of a novel virus such as IDV [19]. On the other hand, lipidomic profiling at multiple time points after infection enables us to describe the processes involved in the induction and resolution of inflammation during the course of infection. Anti-inflammatory drugs that are used to treat BRD act by inhibiting cyclooxygenase (COX) enzymes that oxidize lipids involved in both inflammatory and resolutive pathways. A better characterization of the oxylipid response in the context of infection would improve our understanding of these responses and can ultimately guide both the application of medication and drug development. When used in combination, proteomic and lipidomic techniques can provide a better picture of host–pathogen interactions and disease progression. In this study, we aimed to investigate the early protein response and the kinetics of the lipidomic profile in bronchoalveolar lavage (BAL) supernatants from calves infected with IDV, *M. bovis,* or both, to elucidate the role of IDV as a co-pathogen in BRD.

## 2. Materials and Methods

### 2.1. The Animal Experiment Design and Sample Collection

A total of 29 Normand and Holstein calves were included in an experimental infection study performed at the Research Platform of Infectious Disease (PFIE, National Institute for Agronomic Research, INRAE, Nouzilly, France) in accordance with an ethical agreement (number APAFIS 16364-2018080211232403; French Ministry of Agriculture, Ethics Committee no. 019, approval date October 31th, 2018), as described previously in detail [13]. Briefly, calves were transferred to the facilities when they were between 3 and 6 days old and were demonstrated to be negative for *M. haemolytica*, *P. multocida*, *M. bovis*, *H. somni*, bovine coronavirus, IDV, bovine respiratory syncytial virus, bovine parainfluenza virus type 3, and bovine viral diarrhea virus. The absence of *M. bovis* and IDV-specific serum antibodies was confirmed using ELISA (Bio K 302; BioX diagnostics, Belgium) and hemagglutination inhibition (HI) assays, respectively. The calves were divided into four separate groups according to age and were infected by nebulization with IDV (n = 8), *M. bovis* (n = 8), or IDV and *M. bovis* (n = 8) or were similarly inoculated with non-infected cell culture medium (n = 5). The inoculum consisted of 10^7^ TCID50 of Influenza virus strain D/bovine/France/5920/2014 and/or 10^10^ CFU of *M. bovis* strain RM16 in a 10 mL volume of Dulbecco modified eagle medium (DMEM). Daily clinical examinations were performed, and nasal swabs were collected daily from day 3 pre-infection until 21 days post-infection (−3 to 21 dpi). Blood samples were collected from all calves at −1, 3, 7, 10, 14, and 21 dpi, and bronchoalveolar lavage (BAL) fluid samples were obtained from five calves per group at −5, 2, 7, and 14 dpi, as described previously [13]. Three calves per group, which were sacrificed on day 6, were excluded from the BAL sampling.

The clinical and pathological findings are summarized in the introduction above, and these data were described previously in detail [13]. As additionally previously described, IDV-RNA was detected in all IDV-infected calves, in both nasal swabs and BAL samples, and *M. bovis* DNA was detected in all co-infected calves in nasal swabs. Moreover, *M. bovis* DNA was detected in BAL samples from two out of five single *M. bovis*-infected calves and four out of five co-infected calves. Significantly higher concentrations of *M. bovis* DNA copies were detected in co-infected than in single-infected calves—in nasal secretions 4 dpi and in BAL samples 7 dpi. All non-infected calves were found negative for both pathogens by RT-qPCR in both nasal swabs and BAL samples.

### 2.2. Sample Processing

The BAL samples were filtered through two layers of sterile gauze, and the BAL cells and supernatant were separated by centrifugation, as previously described [20]. The supernatants were stored at −75 °C, prior to mass spectrometric analyses.

### 2.3. Proteomics

Bronchoalveolar lavages collected at 2 dpi were analyzed by mass spectrometry. Prior to analysis, sample volumes were reduced using Speedvac (Thermo Fisher Scientific), and the total protein concentration was measured by the Bradford Protein Assay using bovine serum albumin, as standard. Aliquots containing 7 μg of protein were digested by trypsin overnight at 37 °C after reduction and alkylation. Before MS analysis, the peptides were purified and desalted by Pierce C18 Spin Columns (Thermo Fisher Scientific). These columns were activated by 2 × 200 μL of 50% acetonitrile (ACN) and equilibrated with 2 × 200 μL of 0.5% trifluoroacetic acid (TFA). The tryptic peptides were adsorbed to the media using two repeated cycles of 40 μL sample loading and the column was washed using 3 × 200 μL of 0.5% TFA. Finally, the peptides were eluted in 3 × 50 μL of 70% ACN and dried.

The dried peptides were reconstituted in 50 μL of 0.1% formic acid and diluted threefold for nano-LC-MS/MS analysis. The samples were analyzed using a QExactive Plus Orbitrap mass spectrometer (Thermo Fisher Scientific, Bremen, Germany) equipped with a nano-electrospray ion source. The peptides were separated by reversed-phase liquid chromatography using an EASY-nLC 1000 system (Thermo Fisher Scientific). A set-up of a precolumn and an analytical column was used. The precolumn was a 2 cm EASY-column (1D 100 μm, 5 μm C18) (Thermo Fisher Scientific), while the analytical column was a 10 cm EASY-column (ID 75 μm, 3 μm, C18; Thermo Fisher Scientific). Peptides were eluted with a 90 min linear gradient from 4% to 100% ACN at 250 nL min/L. The mass spectrometer was operated in positive ion mode, acquiring a survey mass spectrum with a resolving power of 70,000 (full-width half maximum), *m*/*z* = 400–1750 using an automatic gain control (AGC) target of 3 × 106. The 10 most intense ions were selected for higher-energy collisional dissociation (HCD) fragmentation (25% normalized collision energy), and MS/MS spectra were generated with an AGC target of 5 × 105 at a resolution of 17,500. The mass spectrometer worked in data-dependent mode.

The acquired data (.RAW files) were processed in MaxQuant version 1.5.3.30, and database searches were performed using the implemented Andromeda search engine. MS/MS spectra were correlated to a FASTA database containing proteins from the Bos taurus proteome, IDV, and *M. bovis*. A decoy search database, including common contaminants and a reverse database, was used to estimate the identification false-discovery rate (FDR). An FDR of 1% was accepted. The search parameters included: maximum 10 ppm and 0.6 Da error tolerances for the survey scan and MS/MS analysis, respectively; the enzyme specificity was trypsin; a maximum of one missed cleavage site was allowed; and cysteine carbamidomethylation was set as static modification and oxidation (M) was set as variable modification. The search criteria for protein identification were set to at least two matching peptides. Label-free quantification was applied for comparative proteomics.

The results from all fractions were combined for a total label-free intensity analysis of each sample. The collected data were filtered to include only proteins that were identified and quantified in at least four calves within at least one of the two groups compared, as previously described [21]. As previously described by Aguilan et al., label-free protein quantities (LFQ) were log2-transformed and normalized by scaling each value against the average of all proteins within a given sample [21]. A probabilistic minimum imputation technique based on the normal distribution was used to address missing values. Proteins with a differential fold change of less than −1 or greater than 1 and a *p*-value of 0.05 (−log2 *p*-value > 4.3219) were further analyzed using Ingenuity Pathway Analysis (IPA) software (version 90348151, 2023 QIAGEN).

### 2.4. Lipidomics

Lipids mediators were analyzed in BAL samples collected at 2, 7 and 14 dpi. The extraction protocol and LC-MS/MS analysis were performed as described in Le Faouder et al. [22]. Briefly, after protein precipitation with methanol, the samples were extracted by solid phase extraction using Oasis-HLB 96-well plates (Waters). Lipids mediators were eluted with methanol and methyl formate. The results were expressed in pg/mL of fluids.

### 2.5. Statistics

Statistical analyses of the lipidomic results were performed using the Kruskal–Wallis test with Dunn’s multiple comparisons. For proteomic data, the normality was assessed using the Shapiro–Wilk test. If both replicates showed a normal distribution, the F-test was used to determine the homogeneity of variance, and the appropriate *t*-test (with equal or unequal variance) was then chosen for statistical comparison. Proteins with a significant deviation from a normal distribution, as determined by the Shapiro–Wilk test, were analyzed using the Mann–Whitney test. All graphs were created by GraphPad 9.5.0 (La Jolla, San Diego, CA, USA).

## 3. Results

### 3.1. IDV Suppresses the Coagulation and the Acute-Phase Response Induced by M. bovis

In total, 1188 proteins were detected in the BAL samples of infected and non-infected calves. The average (range) number of semi-quantified proteins were 480 (321–590), 527 (371–796), 462 (304–618), and 489 (440–545) in controls, IDV-, *M. bovis-*, and IDV+*M. bovis*-infected calves, respectively.

In order to identify the variance of the total protein expression for each animal 2 dpi, a principal component analysis (PCA) was performed. Although individual variability between calves was observed, a tendency of clustering according to treatment was detected as early as 2 dpi, with a similar pattern observed in the control and co-infected calves. (Figure 1).

Next, we examined the differences in protein expression in BAL samples from infected calves and controls and analyzed the probability of the activation or inhibition of biological pathways using IPA. For each comparison, the top-five pathways with the highest *p*-values were illustrated (Figure 2). The pathway named “Role of Tissue Factor in Cancer” was excluded from the analysis comparing *M. bovis*-infected calves and control calves due to the lack of relevance to acute infectious respiratory disease in calves.

Four different comparisons were made to investigate the host response to IDV and *M. bovis* as single infections or the role of IDV and *M. bovis* as co-pathogens: IDV vs. control; *M. bovis* vs. control; IDV+*M. bovis* vs. *M. bovis;* and IDV+*M. bovis* vs. IDV. Overall, a total of eight canonical pathways were identified, which included the coagulation system, intrinsic prothrombin activation, extrinsic prothrombin activation, LXR/RXR activation, FXR/RXR activation, the neutrophil extracellular trap, acute-phase response signaling, and airway pathology in chronic obstructive pulmonary disease. Due to the small number of differentially expressed proteins in IDV+*M. bovis-* vs. IDV-infected calves, no pathways were identified in this comparison.

A total of 59 proteins were differentially expressed between IDV-infected calves and controls. Compared to the controls, the BAL samples of IDV-infected calves contained 22 downregulated proteins and 37 upregulated proteins (Figure 2, Appendix A). Of these, nineteen proteins were found to be involved in the top-five most significant pathways. Among these 19 proteins, 7 were associated with the acute-phase response and 10 were downregulated. The acute-phase response pathway was the most significantly affected pathway, with a −log (*p*-value) of 12.2 (Figure 2). In particular, APOA2 was strongly downregulated by IDV, with a fold change of −5 and a −log 10 (*p*-value) of 6.12. On the other hand, seven proteins were implicated in the neutrophil extracellular trap pathway, six of which were upregulated (Figure 2).

Overall, 25 proteins were differentially expressed between *M. bovis*-infected calves and controls, with 15 proteins downregulated and 10 proteins upregulated by *M. bovis* (Figure 3). Of these, seven proteins were found to be involved in the top-five affected pathways. The coagulation system pathway emerged as the most significantly upregulated pathway, with a −log (*p*-value) of 9.75 (Figure 3). Unlike in IDV-infected calves, the acute-phase response pathway was activated by the upregulation of four proteins: C9, F2, FGB, and FGG).

To investigate the effect of IDV during co-infection with *M. bovis*, we compared the proteome of IDV+*M. bovis*-infected calves and *M. bovis*-infected calves. A total of 58 differentially expressed proteins were detected, of which 17 were downregulated and 41 were upregulated by the addition of IDV (Figure 4). Despite being able to identify thirty-seven of the upregulated proteins, the software was not able to find any of these proteins among the top-five pathways. By performing a manual analysis of each of the upregulated proteins, six proteins (GPI, CD177, OSTF1, S100A11, CTSD, and GSTP1) can be associated with neutrophil degranulation. A total of thirteen downregulated proteins were identified in the five most significantly expressed pathways. The coagulation pathway was observed to be the most significantly affected pathway, with a −log 10 (*p*-value) of 12, similar to *M. bovis*-infected calves when compared to controls. However, whereas this pathway was upregulated by *M. bovis* during a single infection, it appeared downregulated by IDV during co-infection. The acute-phase response signaling pathway contained the highest number of detected proteins, with a total of nine proteins: A2M, AMBP, C4A/C4B, FGA, FGB, FGG, FN1, and PLG. Plasminogen was particularly differentially expressed, with a *p*-value of 0.004, and, similar to FGA, AMBP, and FETUB, was differentially expressed only in the comparison between IDV+*M. bovis*-infected calves vs. *M. bovis*-infected calves (Table 1).

To assess the role of *M. bovis* as a co-pathogen at 2 dpi, we compared the proteome of IDV+*M. bovis*-infected calves with that of IDV-infected calves. Only eight proteins were differentially identified in the analysis, of which four were downregulated (mucin, LPO, cathepsin G, bnbd9, and PIGR) and three were upregulated (APOA1, C5, and AOC3) (Figure 5). None of these proteins was identified when comparing *M. bovis*-infected calves with non-infected calves. As mentioned above, due to the low number of proteins identified, it was not possible to identify canonical pathways.

### 3.2. Co-Infection with IDV and M. bovis Induced a Higher Oxylipid Concentration in BAL Samples Than Either Single Infection at Day 7 Post-Infection

To investigate the processes of inflammation and resolution in more detail, a mass spectrometry (LC-MS/MS) analysis of lipid intermediates and mediators was performed in BAL samples obtained from all calves at 2, 7 and 14 dpi. Based on their polyunsaturated fatty acid (PUFA) precursors, 23 lipids were associated with 3 main oxylipid pathways. The arachidonic acid (ARA) pathway consisted of PGE2, 6K-PGF1A, TXB2, 5-HETE, LTB4, 15-HETE, LXA4, LXB4, and 12-HETE. The docosahexaenoic acid (DHA) pathway included 14-HDOHE, 7(R)-MAR1, MAR2, 17-HDOHE, PDX, PD1, RVD1, RVD2, RVD5, RVD3, and RVD4. Furthermore, the eicosapentaenoic acid (EPA) pathway involved 18-HEPE, RVE1, and RVE2.

At 2 dpi, no significant difference in the activation of any of the oxylipid pathways was detected between the calves in the different groups, but the controls had the lowest overall oxylipid concentration (Figure 6).

At 7 dpi, a high expression of the ARA pathway was detected in the IDV+*M. bovis*-infected calves, followed by IDV-infected calves, *M. bovis*-infected calves, and controls. The increase in lipid expression within the ARA pathway between 2 and 7 dpi was significant only in the IDV+*M. bovis*-infected calves (*p* < 0.01). Between day 2 and 7, the average concentration of the different oxylipids in this pathway increased 8.8-fold in IDV+*M. bovis*-infected calves, 3.2-fold in IDV-infected calves, 1.4-fold in *M. bovis*-infected calves and 2.9-fold in the controls. Notably, among the controls, the increase was mainly attributed to one animal (No. 9238). Despite that, neither IDV nor *M. bovis* was detected in the respiratory tract of this calf, and despite having a normal body temperature and no microscopic lesions in all investigated organs, calf 9238 showed mild clinical signs from 1 dpi, such as decreased appetite, a cough, and abnormal respiratory rate. The removal of calf 9238 from the statistical analysis did not change the final results of any of the assessments throughout the study. This only affected the mean values within the control group and consequently the visualization of the graphs due to changed standard deviations.

Whereas the expression of ARA lipids at 7 dpi was significantly higher in IDV+*M. bovis*-infected calves than in controls (Figure 6), no significant difference was observed between IDV+*M. bovis*-infected and single-infected calves. Nevertheless, a strong statistical difference was observed in the expression of ARA, DHA, and EPA oxylipids at 7 dpi compared to both 2 dpi (ARA, DHA, and EPA) and 14 dpi (ARA and DHA) in double-infected calves (Figure 6).

A non-significant reduction in the concentration of oxylipids was additionally observed in the other infected calves at 14 dpi compared to 7 dpi. The amount of ARA lipids was 9.39-, 1.93-, and 1.2-fold lower at 14 dpi than at 7 dpi for the IDV+*M. bovis*-, IDV-, and *M. bovis*-infected calves, respectively.

In terms of individual lipids, a high production of ARA-oxylipids, including PGE2, 6K-PGF1A, TXB2, 5-HETE, LTB4, 15-HETE, and 12-HETE, and the DHA lipid 14-HDOHE, was observed in all calves throughout the experiment (Figure 7). Most of the individual lipids detected in the infected calves peaked at 7 dpi.

Among the oxylipids analyzed, concentrations of PGE2, TXB2, and LTB4 were statistically different between IDV+*M. bovis*-infected calves and controls at 7 dpi (Figure 8). At 7 dpi, TXB2 was the most abundant lipid in both IDV-infected and IDV+*M. bovis*-infected calves, making it the most abundant oxylipid throughout the study.

## 4. Discussion

This study comprehensively analyzed the bronchoalveolar proteome and lipidome in calves simultaneously infected with IDV, *M. bovis,* or both.

The proteomic data indicated that IDV downregulates acute-phase response signaling and counteracts *M. bovis*-induced coagulation system activation, which are two key mechanisms in the innate immune response. In parallel, high concentrations of oxylipids that derived from the ARA pathway were detected at 7 dpi, particularly in calves infected with both pathogens.

Proteomic analyses were performed at 2 dpi, providing information on the early host response to infection. Of note, IDV, *M. bovis,* or the two pathogens in combination induced significant changes in the expression of several proteins closely associated with innate immunity as early as 2 dpi. In particular, proteins such as A2M, F2, FGA, FGG, FGB, KNG1, PLG, and SERPINC1 were found to be differentially expressed. These proteins are involved in coagulation and the acute-phase response, explaining the recurrent presence of these pathways in the enrichment analyses. While the coagulation pathway was significantly upregulated in response to *M. bovis* infection, as demonstrated by comparing *M. bovis*-infected calves with uninfected controls, the addition of IDV to induce co-infection (IDV+*M. bovis*) resulted in an opposite effect, with the downregulation and counteraction of this response compared to *M. bovis* infection alone.

In the context of IDV vs. controls, the coagulation pathway was not identified among the top-five most significantly differentially regulated pathways; however, A2M and SERPINC1, which are proteins involved in the coagulation pathway, were downregulated with a significant −log10 *p*-value of 1.35 and 1.69, respectively (Appendix A). The proteins associated with the coagulation pathway (F2, FGB, FGG, KNG1, and SERPINC1) in the *M. bovis*-infected calves are predominantly associated with fibrin formation. The formation of fibrin limits infections by supporting immune cells such as neutrophils and macrophages, which are critical for pathogen clearance [23]. However, a study of *Mycoplasma pneumoniae* in children found elevated fibrinogen levels in the most severely affected children, indicating a correlation between coagulation and disease severity. The authors suggested that the excessive coagulation might be related to a massive interleukin secretion causing vascular damage [24]. Our results indicate that *M. bovis* infection induced rapid fibrin formation in the lungs of calves, as a host response. In contrast, the addition of IDV to *M. bovis* resulted in a downregulation of coagulation-associated proteins. Interestingly, with the exception of F2, proteins upregulated by *M. bovis* alone were downregulated by the addition of IDV as a co-infection. This would suggest that IDV reduces the fibrin formation triggered by *M. bovis* early in infection, delaying *M. bovis* clearance and allowing *M. bovis* replication and dissemination. Moreover, although the effect of *M. bovis* as a co-pathogen was not associated with any pathway, the observed downregulation of cathepsin G and LPO was probably related to neutrophil degranulation, which would support the idea that the combination of both pathogens delays the host response. This observation is consistent with the results of the in vivo experiment performed by Lion et al. [13].

Furthermore, A2M, PLG, and FGG were also downregulated in the IDV+*M. bovis* infection compared to *M. bovis* alone. Notably, PLG was additionally downregulated in IDV-infected calves compared to uninfected controls; however, this was with a *p*-value of 0.06 (Appendix A). In Influenza A virus infection, the conversion of plasminogen to plasmin activates the viral haemagglutinin and promotes the replication of the virus. Since plasminogen additionally induces cell infiltration and cytokine production, plasminogen-deficient mice show reduced lung inflammation and damage during Influenza A infection [25]. The observed reduction in PLG associated with IDV, in contrast to what has been described for Influenza A, may be one of the reasons for the low virulence in cattle and apparent non-pathogenicity in humans compared to IAV.

The acute-phase response pathway showed differential activity between infected calves. In particular, there was a downregulation of the acute-phase response when IDV was involved, in contrast to the upregulation observed in *M. bovis* single-infected calves. As previously highlighted, several coagulation proteins also contribute to the acute-phase response. However, with IDV, additional proteins (AGT, APOA2, C5, FN1, HRG, ITIH2, SERPIND1, AMBP, and C4A/B) were identified as downregulated, emphasizing the role of this pathway. The acute-phase response is responsible for the early initiation of several defense mechanisms against infectious diseases, such as inflammation, activation of immune cells, or fever [26]. Particularly, C4A/B was significantly downregulated in both IDV single- and co-infected calves, compared to uninfected controls and *M. bovis* single-infected calves, respectively. Moreover, C4, a key protein in the acute-phase response, plays an important role in both the classical and lectin complement pathways. Downregulation of C4 can lead to impaired processes such as opsonization, clearance of immune complexes, inflammatory responses, and cell recruitment, increasing susceptibility to secondary infections [27]. In parallel with the counteracting effect described for the fibrin-formation process, IDV seems to reduce the activation of basic innate immune processes. The reduced concentrations of C4 could be a key mechanism, facilitating the replication of *M. bovis*.

These results are also in line with a study by Gaudino et al., who used precision-cut lung slices as an organotypic lung model to demonstrate how IDV impairs the innate immune response induced by *M. bovis*. This impairment was characterized by a decrease in the expression of pro-inflammatory cytokines and chemokines, such as IL-8, IL1β, and IL-17 [18].

Lipid mediator profiling of infected calves suggested the generation of several key oxylipids, produced through catalyzation by COX-1, COX-2, and LOX, across the different infection protocols. The highest total oxylipid expression was observed at 7 dpi in double-infected calves and was characterized by oxylipids that derived from arachidonic acid. Although the high ARA lipid concentrations at 7 dpi coincided with the peak of clinical signs on day 8.6 ± 1.3 in the IDV+*M. bovis*-infected calves at the group level, the individual calves with the highest oxylipid concentrations in BAL samples did not have the most pronounced clinical signs (data not shown) [13]. It should be noted that the lipidomic profile may vary depending on the viral strain studied. For example, Tam et al. demonstrated different lipid expression patterns between high and low pathogenic strains of Influenza A virus infection in mice [28]. This highlights the need to study a wider range of IDV strains in the future.

At 2 dpi, the detection of PGE2 was significantly higher in the IDV+*M. bovis*-infected calves than in controls (*p*-value < 0.05). Prostaglandin E2 plays a dual role in inflammation. On one hand, it is characterized by its vasodilatory properties and ability to attract macrophages to the site of inflammation [29,30]. On the other hand, it induces a switch to the resolution of inflammation by reducing the infiltration of neutrophils, promoting the production of anti-inflammatory cytokines such as interleukin-10, and inducing both apoptosis and the efferocytosis of neutrophils [31,32,33].

In contrast to 2 dpi, which was mainly characterized by PGE2 responses, additional lipids such as TXB and LTB4 were produced in higher amounts, particularly in the IDV+*M. bovis*-infected calves, at 7 dpi. The activation of COX-1 (TXB), COX-2 (PGE2), and LOX-5 (LTB4) in doubly infected calves on day 7 compared to uninfected controls suggests that the use of selective inhibition drugs of a single pathway may not entirely suppress inflammation. Thromboxane is involved in platelet aggregation and has vasoconstrictive properties, whereas LTB4 has a pro-inflammatory role in neutrophil recruitment, which influences immune defense mechanisms against viral infections [34,35]. A daily administration of LTB4 to IAV-infected mice enhanced the reduction of lung viral loads through the upregulation of antimicrobial peptides, compared to mice treated with a placebo [36].

Although no significant difference was found in RVD1 at 7 dpi, some of the IDV+*M. Bovis*-infected calves showed a high concentration of this pro-resolving and anti-inflammatory lipid, suggesting that a process of healing and tissue repair was carried out. In agreement with this, a significant decrease in oxylipids was observed on day 14, suggesting a significant reduction in the inflammatory process.

To fully evaluate the potential of IDV, different pathogen combinations need to be tested, including different infection orders and intervals, as well as pathogen loads. Simultaneous co-infection is a relevant starting point because calves from different herds are often grouped at markets or stocker farms, resulting in rapid and simultaneous exposure to different pathogens.

The proteomic database for cattle likely does not have the same quality and completeness as that for humans or other commonly studied model organisms. However, by comparing infected and uninfected calves, the differential protein expression provides relevant information. By including lung tissue in future investigations, additional information will be collected. The use of BAL samples presents certain challenges, including the risk of dilution effects. However, the possibility of sampling the same individual repeatedly over time, without the need to sacrifice the animal or perform surgery, is a great advantage. Bronchoalveolar lavage also provides a more representative view of the lung’s immune state than analyzing a small sample of tissue, which may over- or under-represent the inflammatory state dependinsg on the area sampled.

In conclusion, this study investigated the BAL proteome and lipidome affected by IDV and *M. bovis* infection in cattle. Our observations suggest that during infection with IDV, certain pathways of the innate immune response, particularly those related to the acute phase and coagulation, are initially suppressed. This early suppression appears to counteract the activation of the host response to *M. bovis* and consequently the clearance of this pathogen during co-infection. In contrast to single infections, co-infections with IDV and *M. bovis* acted synergistically and induced detectable changes in lipid mediators in BAL samples, which were mainly generated by the COX-1, COX-2, and LOX-5-oxidation of arachidonic acid. Although the peak of these responses matched the peak of clinical signs, there was no association between oxylipid concentrations and disease severity in individual calves. Further studies are required to improve our understanding of the role of the identified proteins and oxylipids during Influenza D detection. Further studies with different IDV strains are needed to validate the current findings and to contribute to a more comprehensive understanding of the role of IDV in the pathogenesis of BRD.

## Figures and Tables

**Figure 1 viruses-16-00361-f001:**
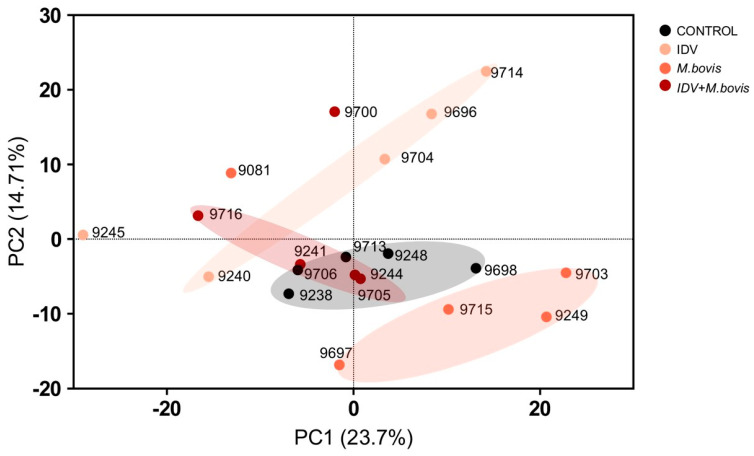
Proteomic principal component analysis (PCA) plot illustrating the distribution of bronchoalveolar proteome in calves at 2 dpi. The black, pink, orange, and red dots correspond to the BAL samples of the controls, IDV−, *M. bovis*− and IDV+*M. bovis*-infected calves, respectively, and are labeled with the calf identity number.

**Figure 2 viruses-16-00361-f002:**
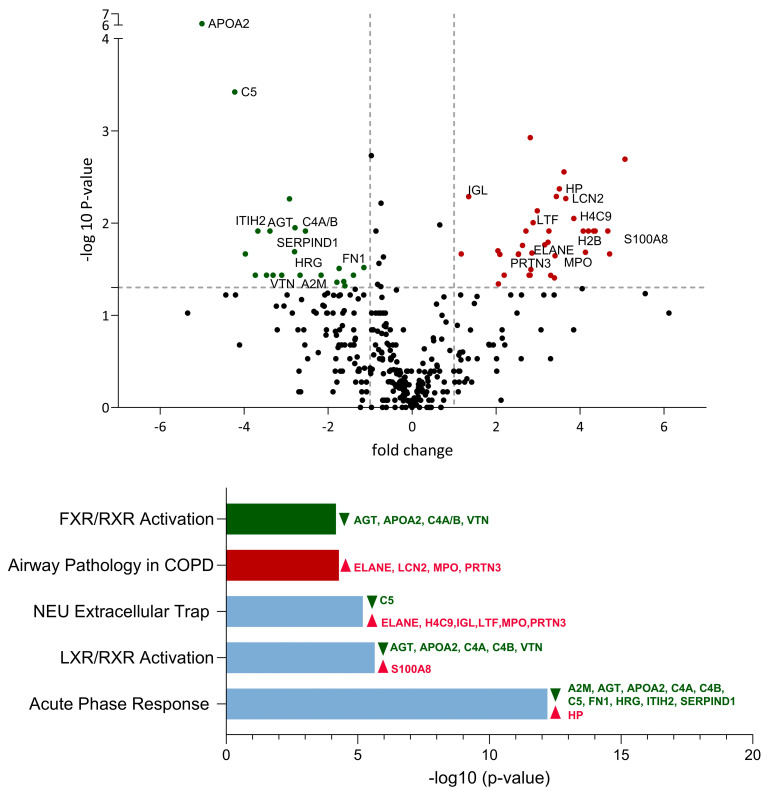
Comparative analysis of protein expression and associated biological pathways in bronchoalveolar lavage of IDV-infected calves vs. controls. Upregulated proteins and pathways are illustrated in red, whereas downregulated proteins and pathways are represented in green. Pathways exhibiting both upregulated and downregulated proteins are indicated in light blue. The proteins involved in the top-five significantly affected pathways are identified by their gene name. All illustrated proteins are listed in the Appendix A.

**Figure 3 viruses-16-00361-f003:**
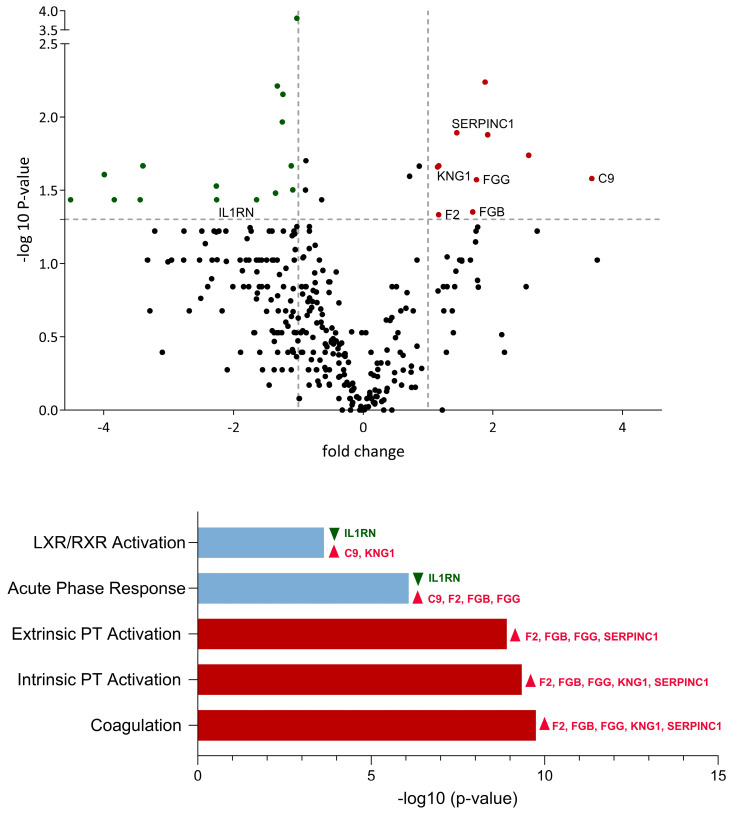
Comparative analysis of protein expression and associated biological pathways in bronchoalveolar lavage of *M. bovis*-infected calves vs. controls. Upregulated proteins and pathways are illustrated in red, whereas downregulated proteins and pathways are represented in green. Pathways exhibiting both upregulated and downregulated proteins are indicated in light blue. The proteins involved in the top-five significantly affected pathways are named with their gene name. All illustrated proteins are listed in the Appendix A.

**Figure 4 viruses-16-00361-f004:**
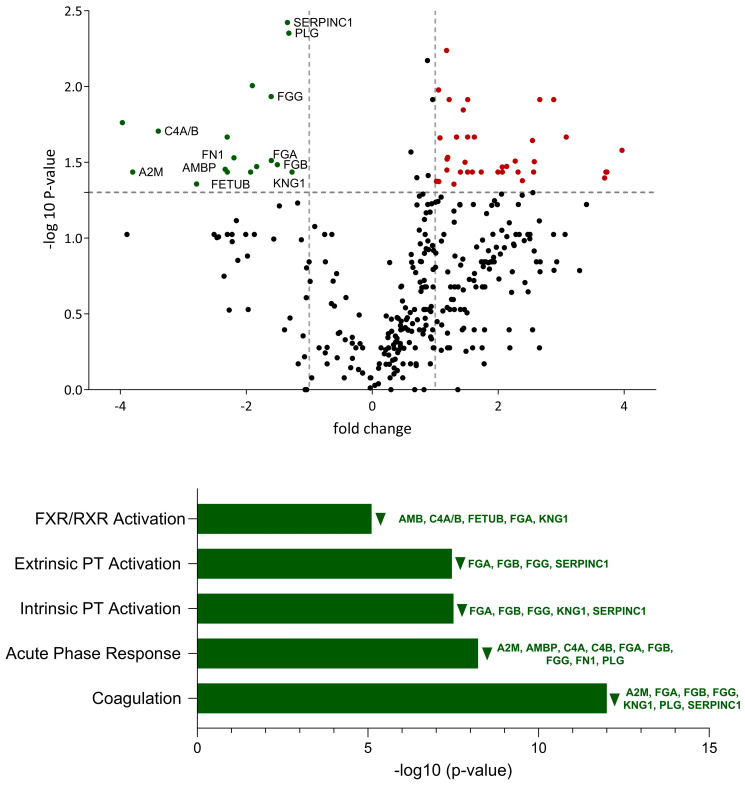
Comparative analysis of protein expression and associated biological pathways in bronchoalveolar lavage of IDV+*M. bovis*-infected calves vs. *M. bovis*-infected calves. Downregulated proteins and pathways are represented in green. The proteins involved in the top-five significantly affected pathways are named with their gene name. All illustrated proteins are listed in the Appendix A.

**Figure 5 viruses-16-00361-f005:**
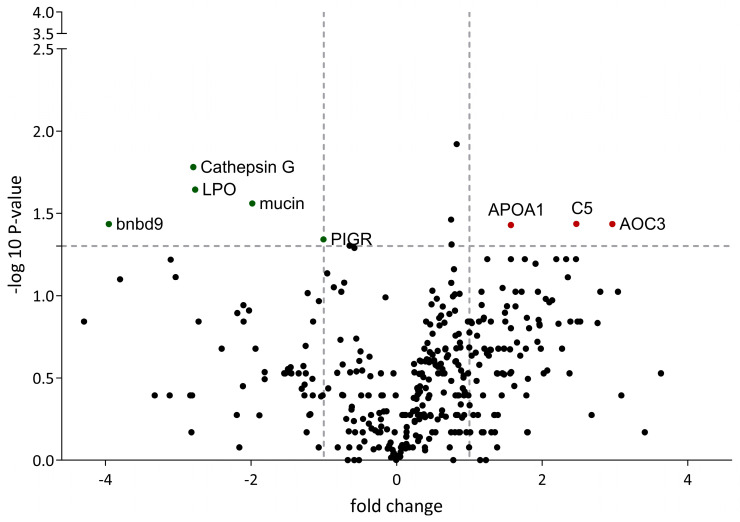
Comparative analysis of protein expression in bronchoalveolar lavage of IDV+*M. bovis*-infected calves vs. IDV-infected calves. Upregulated proteins are illustrated in red, whereas downregulated proteins are represented in green. All illustrated proteins are listed in the Appendix A.

**Figure 6 viruses-16-00361-f006:**
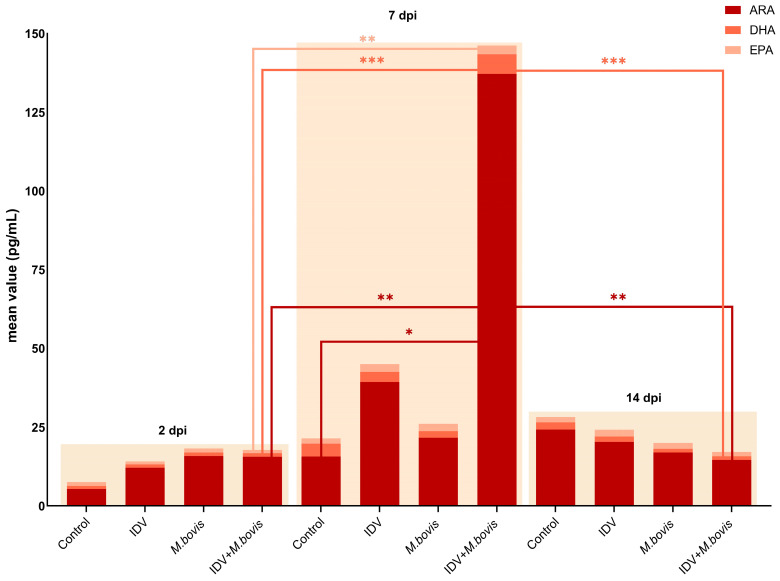
Dynamic changes in lipid oxidation in bronchoalveolar lavage from calves infected with IDV, *M. bovis* or IDV+*M. bovis*, or from uninfected controls, at 2, 7 and 14 days post-infection (dpi). ARA: Arachidonic Acid, DHA: Docosahexaenoic Acid, EPA: Eicosapentanoic Acid. Significance levels are indicated as follows: * (*p*-value < 0.05), ** (*p*-value < 0.01), and *** (*p*-value < 0.001).

**Figure 7 viruses-16-00361-f007:**
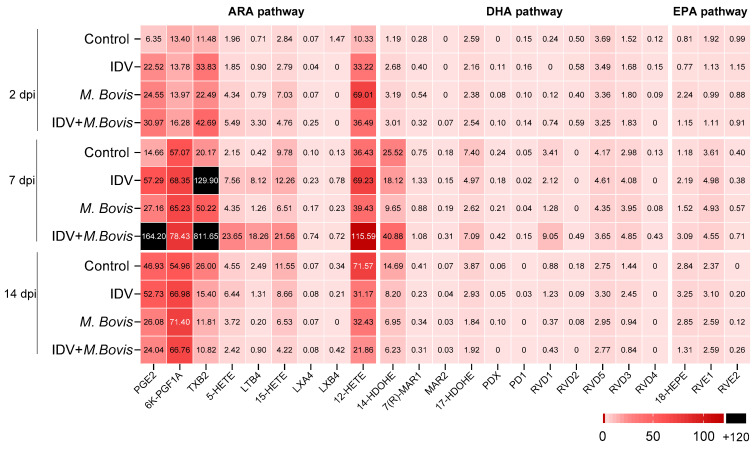
Heatmap displaying the mean concentration (pg/mL) of each of the 23 oxylipids analyzed.

**Figure 8 viruses-16-00361-f008:**
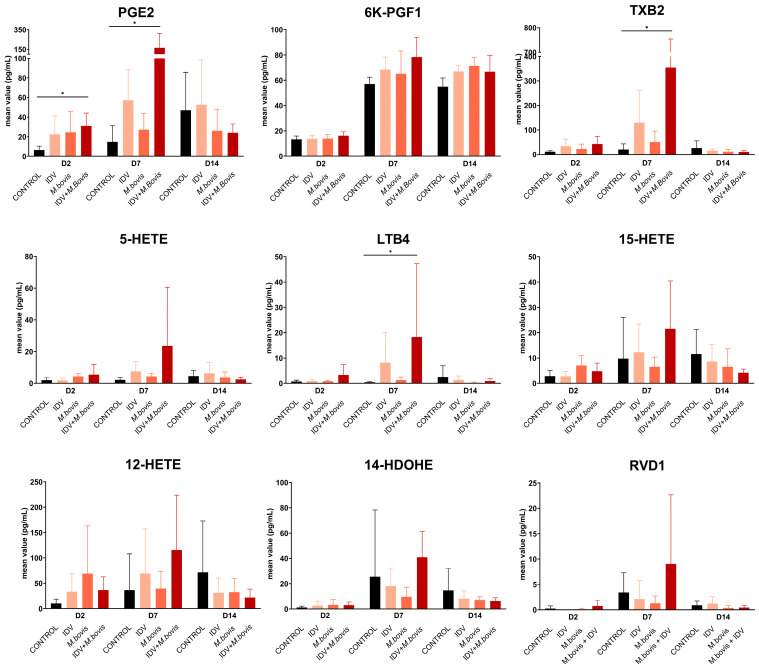
Oxylipids in bronchoalveolar lavage from calves infected with IDV, *M. bovis* or IDV+*M. bovis*, or from uninfected controls, at 2, 7 and 14 days post-infection. The bars denote mean value ± standard deviations. The asterisk indicates significant changes (*p*-value < 0.05).

**Table 1 viruses-16-00361-t001:** Proteins that were differentially expressed only by the addition of IDV to the *M. bovis* infection.

Protein	Description	Fold Change	*p*-Value	Biological Process
FGA	Fibrinogen alpha chain	−1.6	0.03	acute-phase response
PLG	Plasminogen	−1.3	0.004	blood coagulation
AMBP	Alpha-1-microglobulin	−2.29	0.03	negative regulation of immune response
FETUB	Fetuin B	−2.3	0.03	response to systemic inflammation

## Data Availability

Data are contained within the article and Appendix A.

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
