# Peer review of "Proteomic and Lipidomic Profiling of Calves Experimentally Co-Infected with Influenza D Virus and Mycoplasma bovis: Insights into the Host–Pathogen Interactions"

_viruses, 2024, doi:10.3390/v16030361_

Round 1

Reviewer 1 Report

Comments and Suggestions for Authors

The reviewer acknowledges the challenges with large animal infection studies and would like to commend authors for continuing their large animal studies/ co-infection studies. Here are some minor comments related to this paper:

I felt that author were short on including relevant citations on existing IDV  and BRD studies. 

Reference 3 and Reference 15 are same.

Line 63-64: reference?

Line 29: Total animal (n=29), however just few lines below, 4 groups with n=8. Provide correct number of animals.

For IDV+M. bovis infected group: total 20 mL volume was inoculated or 10 mL? That line is confusing and can be elaborated to specify that final volume in co-infection was also 10 mL.

For co-infection model, it is generally regarded that certain viral infection create favorable environment for bacterial colonization. In this regard, does simultaneous co-infection of IDV and M. bovis is representative model? Why not start with IDV challenge followed by M. bovis infection (2-3 days later)?

The IDV-RNA detected in calves however it is unclear whether intact replication competent virus was recovered (active replication vs viral genome only?)

Why only day 2 for proteomics compared to 3 different time point for lipidomics? Cost or any other reason.

How well defined is proteomic database for healthy cattle (especially from BAL)? Can there be bias for detection of extracellular proteins in BAL compared to intracellular proteins (especially when sample volumes are less)?  Should be discussed in discussion section.

Did any of these proteomic changes correlated with pathological changes in lung tissue? Is there a way to perform these studies since authors already have lung tissues from these animals.

Just outside the scope of this paper however, it would be interesting to see if there are any IDV lineage specific differences in co-infection model (D/OK vs D/CA vs D/660) especially in future experiments.

Reviewer 2 Report

Comments and Suggestions for Authors

In the manuscript by Alvarez et al. is the proteomic and lipidomic profiling described of broncheo alveolar lavages (BALs) from calves that were experimentally infected with Influenza D virus (IDV), Mycoplasma Bovis (M. Bovis) or co-infected with both. During this analysis was observed that infection with IDV resulted in the higher presence of proteins that are involved with acute phase of infection (in comparison with healthy calves). In the BALs of calves infected with M. Bovis were primarily proteins upregulated that have an influence on the coagulation pathway. Interestingly, in BALs from co-infected calves were pathways downregulated that have an influence on the coagulation pathway, indicating a dampening effect by IDV, which might contribute to a more successful infection by M. Bovis. In addition, analysis into oxylipids was performed and showed that proteins involved in the ARA pathway are primarily upregulated during co-infection 7 days post-infection.

The results are potentially interesting to unravel the heterogeneric clinical manifestation of IDV and M. Bovis infection in calves and shows the synergy between bacteria and viruses of the bovine respiratory disease complex. However, this reviewer also has major concerns about the lengthy manuscript in its current form and would recommend major revisions.

Major revisions

-      The results from chapter 3.1 are observationally described and easy to understand. However, it is unclear to this reviewer why the analysis in oxylipids described in chapter 3.2 is performed and, in addition, it comes a bit out of the blue since it is only marginally introduced earlier in the manuscript. The first paragraph of chapter 3.2 states the three lipid pathways and which proteins are involved in them, but it is not explained what the role of each of these pathways is. And what is the additional value to look into lipids? More introduction into the lipids is required including their potential role in infection.

-      Figure 1 shows the PCA plot that illustrates the distribution of the proteomes, however, there is only a nice clustering of the uninfected, control group according to the reviewer. The IDV, M. Bovis and co-infected animals are quite dispersed over the plot although the authors claim that the calves clustered according to their group. Especially the IDV infected calves are quite dispersed over the entire plot. What do the authors want to say with this plot? And how does this analysis affect the further analysis?

-      In chapter 3.2 is described that at 7 DPI a high expression of the ARA pathway was observed in the calves. It is very much appreciated by the reviewer that the authors mentioned that one animal (No. 9238) potentially was infected with a (unknown) pathogen, although all calves were pre-screened for other agents of the bovine respiratory disease complex. However, the reviewer wonders to what extend this calf skews the results if the data of this calf was left out of the analysis.

-      In the comparative analysis in figure 4 between the co-infected calves and the M. Bovis-infected calves was mainly observed that proteins involved in the coagulation pathway were downregulated. Unfortunately, the authors did not discuss which upregulated proteins were measured which can be potentially interesting as well. Are proteins involved in acute infection upregulated during co-infection? In addition, is there a specific reason why the comparison between the co-infected and IDV-infected calves was not performed?

Minor revisions

-      Although the majority of the text is clearly described, the readability of certain paragraphs is impaired due to the description of all the proteins and their abbreviation. For example: Line 241-245 and 262-271. Please reread the manuscript and assess if the paragraphs ‘flow’ nicely.

-      Change the times into folds throughout the manuscript. Example: Line 280 is currently: ‘3.2 times in IDV-infected calves, 1.4 times in M-bovis ….’. Suggestion: ‘3.2-fold in IDV-infected calves, 1.4-fold in M.-bovis…’

-      Please included superscript and subscript throughout the manuscript. Line 88-89: Change 10^7 to 107 and 10^10 to 1010

-      Line 256: higer should be higher

-      Figure 5: The interpretation of this bar graph is quite difficult. I would suggest to make the figure in 3 separate bar graphs, so per days post infection.

-      Figure 6 and 7: These 2 figures show the same (mean) data. Please choose one of the figures to state your conclusions.

-      Discussion: is very lengthy and sometimes this reviewer does not understand why a certain paragraph is included and raises the question whether it has any additional value. For instance, Line 402-410 and 411-420. Please revise the discussion and shorten this section.

Comments on the Quality of English Language

The quality of English is sufficient, but the readability is sometimes impaired due to the names of the proteins and their abbreviations.

Round 2

Reviewer 2 Report

Comments and Suggestions for Authors

Thank you for providing adequete answers to the questions.